

# Characterizing Tundra snow sub-pixel variability to improve brightness temperature estimation in satellite SWE retrievals

Julien Meloche[1,2], Alexandre Langlois[1,2], Nick Rutter[3], Alain Royer [1,2], Josh King[4], Branden Walker[5]

[1]Centre d'Applications et de Recherche en Télédétection, Université de Sherbrooke, Sherbrooke, J1K 2R1, Canada
[2]Centre d'études Nordiques, Université Laval, Québec, G1V 0A6, Canada
[3]Department of Geography and Environmental Sciences, Northumbria University, Newcastle upon Tyne, NE1 8ST, UK
[4]Environment and Climate Change Canada, Climate Research Division, Toronto, M3H 5T4, Canada
[5] Cold Regions Research Centre, Wilfrid Laurier University, Waterloo, N2L 3C5, Canada

*Corresponding author*: Julien Meloche (julien.meloche@usherbrooke.ca)

**Abstract.**

Topography and vegetation play a major role in sub-pixel variability of Arctic snowpack properties, but are not considered in current passive microwave (PMW) satellite SWE retrievals. Simulation of sub-pixel variability of snow properties is also problematic when downscaling snow and climate models. In this study, we simplified observed variability of snowpack properties (depth, density, microstructure) in a two-layer model with mean values and distributions of two multi-year tundra dataset so they could be incorporated in SWE retrieval schemes. Spatial variation of snow depth was parametrized by a log-15 normal distribution with mean ($\mu_{sd}$) values and coefficients of variation ($CV_{sd}$). Snow depth variability ($CV_{sd}$) was found to increase as a function of the area measured by a Remotely Piloted Aircraft System (RPAS). Distributions of snow specific area (SSA) and density were found for the wind slab (WS) and depth hoar (DH) layers. The mean depth hoar fraction (DHF) was found to be higher in Trail Valley Creek (TVC) than Cambridge Bay (CB) where TVC is at a lower latitude with a sub-arctic 20 shrub tundra compared to CB which is a graminoid tundra. DHF were fitted with a gaussian process and predicted from snow depth. Simulations of brightness temperatures using the Snow Microwave Radiative Transfer (SMRT) model incorporating snow depth and DHF variation were evaluated with measurements from the Special Sensor Microwave/Imager and Sounder (SSMIS) sensor. Variation in snow depth ($CV_{sd}$) is proposed as an effective parameter to account for sub-pixel variability in PMW emission, improving simulation by 8K. Snow depth simulations using a $CV_{sd}$ of 0.9 best matched $CV_{sd}$ observations 25 from spatial datasets for areas > 3 km², which is comparable to the 3.125 km pixel size of the Equal-Area Scalable Earth (EASE) grid 2.0 enhanced resolution at 37 GHz.

**Keywords**: sub-pixel variability, microwave, snow water equivalent, tundra snow



## 1 Introduction

Snow cover is known to be highly variable at the local scale (10 – 1000 m) due to wind redistribution, sublimation (Liston and
Sturm, 1998; Winstral et al., 2013) and vegetation trapping (Sturm et al., 2001). Physical properties of snow such as
measurement of stratigraphy (Fierz et al., 2009) can be aggregated into layers, but their spatial distribution is highly variable
given their dependence on total depth and surface roughness (Liljedahl et al., 2016; Rutter et al., 2014). Such variability leads
to uncertainties in the retrievals of snow state variables such as snow water equivalent (SWE) using microwave remote sensing

from local scales (King et al., 2018; Rutter et al., 2019) to global scales (Pulliainen et al., 2020). Improving our empirical
understanding of the processes governing this variability would improve space-borne snow monitoring, especially in Arctic
regions where ground measurements and weather station networks are sparse.

Measurement of SWE using passive microwave satellite data (Larue et al., 2018; Pulliainen, 2006) is possible using a radiative
transfer model to simulate snow emission at various frequencies, from which an inversion of the model can produce global

estimates of snow depth (Takala et al., 2011). More specifically, passive microwave brightness temperatures ($T_B$) are governed
by dielectric properties of the layered snowpack. As such, each layer has its own absorption and scattering properties; the
amount of scattering is proportional to snow total mass where the scattering and emission is frequency-dependent (Kelly et al.,
2003). Scattering at higher frequencies such as 37GHz, will lead to lower $T_B$ so differences between $T_B$ at two frequencies (37-
19 GHz) is related to snow mass (Chang et al., 1982). Arctic snowpack mainly consists of two distinct layers (wind slab and

depth hoar), where each layer has unique scattering properties (Derksen et al., 2010). Complexity of the layered properties
(density, temperature and microstructure) strongly influence radiative transfer modelling (King et al., 2015; Rutter et al., 2014).
Furthermore, recent developments in radiative transfer modelling (SMRT: Picard et al., 2018, DMRT: Tsang et al., 2000 and
MEMLS: Wiesmann and Mätzler, 1999), microstructure representation (Royer et al., 2017), and in situ measurement of
snowpack properties (Gallet et al., 2009; Montpetit et al., 2012; Proksch et al., 2015) have provided significant agreement

between models and in situ measurements. However, spatial distribution and heterogeneity of total snow depth and stratigraphy
remains challenging to implement and is not considered for large scale monitoring of SWE in tundra environments. Rutter et
al. (2019) and Saberi et al. (2020), using three- and two-layer models respectively, demonstrated a relationship between the
ratio of depth hoar and wind slab with respect to total depth, enabling the usage of proportion of these two layers with total
snow depth. Working with a simplified layer representation of a snowpack with well-defined physical properties may

adequately characterize snowpack for large scale SWE retrievals.

Two dominant processes governing snow depth variability in the Arctic are 1) wind redistribution with topography (Sturm and
Wagner, 2010; Winstral et al., 2002) and 2) vegetation trapping (Domine et al., 2018; Sturm et al., 2001). Liston (2004)
described snow depth heterogeneity using a log-normal distribution with a coefficient of variation of snow depth ($CV_{sd}$), the
ratio between standard deviation ($\sigma_{sd}$) and the mean of snow depth ($\mu_{sd}$), indicating the extent and spread of a distribution

(i.e. high variability over thin snow will lead to high values of $CV_{sd}$). Also, Liston (2004) proposed 9 categories of $CV_{sd}$ with





values ranging from 0.9 to 0.06 for mid-latitude treeless mountains to ephemeral snow, where arctic tundra type was 0.4. Snow depth variability is based on a parametrization of $\mu_{sd}$, $CV_{sd}$ on the log-normal distribution scale parameters (λ, ζ). Gisnas et al. (2016) adapted that approach using scale parameters (α, β) of the gamma distribution. In all cases, $CV_{sd}$ is used to describe subgrid variability (Clark et al., 2011), but its value remains challenging to quantify given that regional trends are linked to

topography, vegetation and climate (Winstral and Marks, 2014). In this context, $CV_{sd}$ is used to quantify spatial heterogeneity of snow in climate modelling, but so far has not been used in microwave SWE retrievals.

In SWE retrievals, snow depth is assumed to be uniform and the mean depth is used to optimize brightness temperature and derive SWE from depth and assumed density (Kelly, 2009). There is potential for $CV_{sd}$ to be used as an effective parameter to estimate sub-pixel variability in brightness temperature. Bayesian frameworks are used in inversion schemes for SWE

retrievals (Durand and Liu, 2012; Pan et al., 2017; Saberi et al., 2020) using *a priori* information (density, microstructure and temperature) from regional snowpack characteristics and inversion of radiative transfer models (Saberi et al., 2020). An iterative approach based on Bayesian theory is used (Takala et al., 2011) to match observed brightness temperature with modelled brightness temperature by iterating *a priori* information of the snowpack in order to derive snow depth and SWE. Saberi et al. (2020) conducted a case study for snow depth retrievals using a two layer model from airborne microwave

observations using a Bayesian framework (or Monte Carlo Markov Chain) over tundra snow. However, high uncertainty (21.8 cm) in retrieved snow depth (via $T_B$) resulted, which suggested the use of a Gaussian Process (GP) involving snow depth instead of a uniform snow depth.

To address this research gap, we used a multi-year snow dataset from two Arctic locations to quantify sub-pixel variability of snow depth and microstructure and used $CV_{sd}$ as an effective parameter that controls snow sub-pixel variability. Firstly, we

evaluate tundra snow depth spatial variability using probability density functions (log-normal and gamma) and its parameters, $\mu_{sd}$ and $CV_{sd}$. Secondly, we present distinct snow microstructure and density values of both tundra main layers (depth hoar and wind slab), mean ratios of layer thickness and their properties relative to snow depth. Finally, we perform a Gaussian process fit to estimate depth hoar fraction (DHF) from snow depth, using probability density functions of snow depth. Then we compare mean pixel snow properties with simulations of sub-pixel variation in snow properties to evaluate biases between

measured $T_B$ from a satellite sensor at 37 GHz, and $T_B$ simulated by inversion of a radiative transfer model.

## 2 Methods

### 2.1 Study site

Data were collected in two regions of the Canadian Arctic, with different topography and vegetation yielding different snow depth distributions. Trail Valley Creek (TVC) research watershed, Northwest Territories (68°44' N, 133°33' W), located at

the southern edge of arctic shrub-tundra, is dominated by herbaceous tundra and dwarf shrubs and characterized by gently





rolling hills with steep slopes. Greiner Lake watershed, Cambridge Bay (CB), Nunavut (69°13' N, 104°53' W), located within arctic tundra, is characterized by dwarf shrub and calcareous tills on upland sites with gently rolling hills and small ponds and lakes. TVC is considered to have more sub-arctic attributes with predominant vegetation than CB given its proximity to the Northern edge of the boreal forest. Topographic maps (Figure 1; ArcticDEM), show slightly higher variation in elevation at

TVC with plateau and steep slopes compared to CB which is dominated by ponds and small variation in topography.

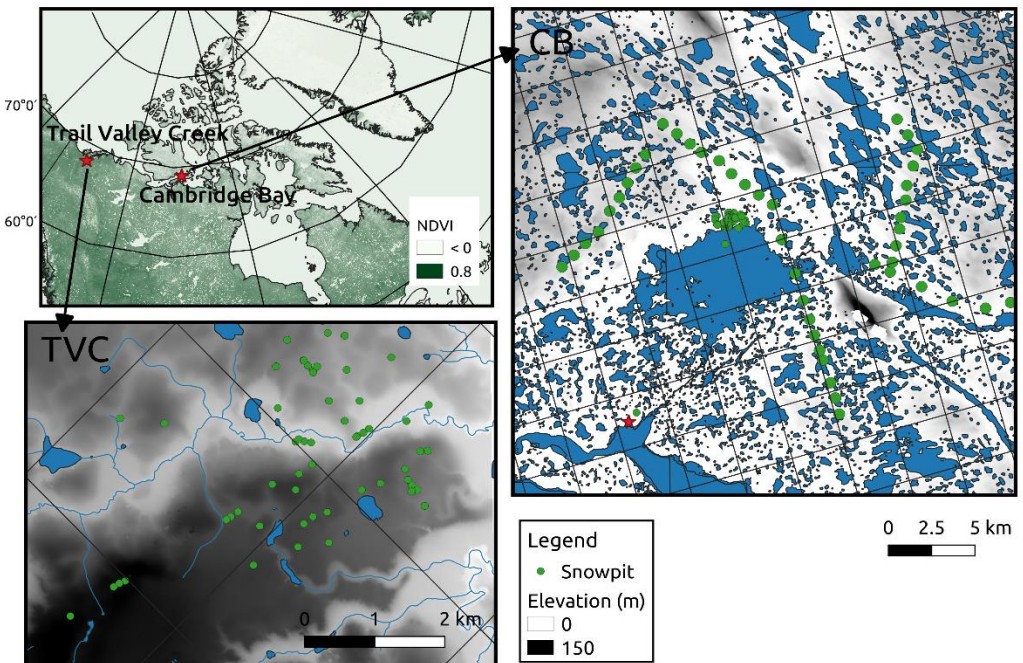

**Figure 1: Locations of study areas in the Canadian Arctic, Cambridge Bay and Trail Valley Creek site. Grid shown is the enhanced 3.125 km EASE grid 2.0 used for satellite data. The ArcticDEM is a 2 m-resolution (Morin et al., 2016) derived from stereo high-resolution visible imagery for the entire Arctic domain, freely available.**

**2.2 Data**

Snow pits (315) at each site (TVC: 68, CB: 248) provided information on snow layering, vertical profiles of geophysical properties (includes temperature, grain type classification, hardness, density, microstructure, and depth). Measurements of visual stratigraphy and grain type classification was conducted following Fierz et al. (2009). Density was measured using 100 cm$^3$ density cutters and digital scales. Snow specific surface area (SSA) was measured using an InfraRed Integrating Sphere

(IRIS) (Montpetit et al., 2012b) in Cambridge Bay, and an A2 Photonic Sensors IceCube in TVC, both based on 1300 nm laser reflectometry (Gallet et al., 2009). Snow depth measurements, linear transects and circular transect around snow pits, used a magnaprobe from SnowHydro LLC (Sturm and Holmgren, 2018), which is equipped with a standard GPS unit. Measured





snow depth distributions were used to identify subsequent pit locations (on site) from a predefined transect across CB watershed in order to ensure the snow pit locations were representative of wider spatial variability (Table 1). For TVC, pit

locations were chosen based on previous snow depth distribution (2016), slope and elevation. Multiple snow depth maps at 1m resolution from RPAS surveys conducted in March 2018 (Walker et al., 2020) were used to estimate snow depth distribution in TVC with total spatial coverage of 13 km². Also, a small RPAS survey is available for CB with spatial coverage of 0.2 km² at 1 m resolution. Maps of normalized difference vegetation index (NDVI) were created from Sentinel-2 (10 m resolution) images from late summer (2019-09-01 for TVC and 2019-09-08 for CB).


**Table 1: Summary of number of snow depth measurements (Magnaprobe and RPAS) and snow pit sites per year. The availability of SSA and density measurements across sites and years are also noted (x). See Table 2 for full dates.**

| Site | Date | Magnaprobe | Snowpit | SSA | Density |
|---|---|---|---|---|---|
| TVC | March 15 -25, 2019 | 8541 | 32 | x | x |
| | March 15 -23, 2018 | 7190 | 36 | x | x |
| TVC-RPAS | March 12- April 22, 2018 | Pixels : 6 325 365 Resolution : 1m | | | |
| CB-RPAS | April 15, 2018 | Pixels : 72 902 Resolution : 1m | | | |
| CB | April 15-29, 2019 | 982 | 64 | x | x |
| | April 12-24, 2018 | - | 50 | x | x |
| | May 1-8, 2017 | 4045 | 51 | | x |
| | April 2-10, 2016 | 3403 | 35 | | x |
| | April 9-16, 2015 | 12 282 | 48 | | x |

## 120 2.3 Measured brightness temperatures and Snow Microwave Radiative Transfer (SMRT)

Microwave $T_B$ were used to evaluate simulations from SMRT at 37 GHz from the Special Sensor Microwave/Imager and Sounder (SSMIS) sensor, EASE 2.0 grid resampled at 3.125 km resolution (Brodzik et al., 2018), for both TVC and CB regions. $T_B$ were spatially averaged to match snow pit area   (CB : 24 pixels, TVC : 14 pixels) and filtered to remove any contribution from sea or deep lakes, as pixels with liquid water exhibit large biases even if the signal at 37 GHz is mostly

sensitive to snow (Derksen et al., 2012). $T_B$ were temporally averaged to match times of field measurements, representing peak winter snow accumulation (Table 2). Also, $T_B$ were corrected for atmospheric contributions using the linear relation with precipitable water from the 29 atmospheric NARR layers (Vargel et al., 2020; Roy et al., 2013).

A multi-layered snowpack radiative transfer model (SMRT,  Picard et al., 2018) was used to simulate snow emission at 37

GHz. Model inputs are snow temperature, density and microstructure of each snow layer. Correlation length of snow





microstructure in each layer was estimated from mean density and SSA measurements of each layer (WS and DH) using Debye's equation scaled by a factor ($\kappa = 1.39$) for arctic snow as suggested by Eq. (3b) and (4) in Vargel et al. (2020) with the Improved Born Approximation (IBA-Exp) configuration. Soil emission was simulated using the Wegmüller and Mätzler (1999) model with permittivity and roughness values from a field study of frozen soil emission based in CB (Meloche et al.,
2020). The soil parameters from CB (Meloche et al., 2020) closely match values from a study in TVC (King et al., 2018) and were used for both sites simulation. The basal layer temperature was set to the mean soil-DH interface measurements from snow pits of each site. The temperature of the WS layer was estimated from the North American Regional Reanalysis (NARR) air surface temperature, which closely matched snow pit surface layer temperature. NARR air surface temperatures were used because it provides a global estimate that matches spatial coverage of the EASE grid, which is continuous (spatially and
temporally) compared to the sparse snow pit observations.

**Table 2: Summary of mean basal and air surface temperatures for SMRT simulations, precipitable water (PWAT) used for atmospheric correction and measured (corrected) $T_B$ at both polarization vertical (V) and horizontal (H) by the SSMIS sensor (platform F18).**

| Sites | $T_{base}$ (K) | $T_{surface}$ NARR (K) | PWAT (mm) | $T_B$ H (K) | $T_B$ V (K) |
|---|---|---|---|---|---|
| CB (April 15-29, 2019) | 257 | 261.5 | 3.61 | 195.3 | 211.0 |
| CB (April 12-24, 2018) | 257 | 260.1 | 3.72 | 179.3 | 195.7 |
| CB (May 1-8, 2017) | 263 | 261.3 | 3.33 | 187.1 | 205.0 |
| CB (April 2-10, 2016) | 256 | 258.8 | 2.80 | 190.1 | 215.4 |
| CB (April 9-16, 2015) | 254 | 256.2 | 2.34 | 193.0 | 215.9 |
| TVC (March 15 -25, 2019) | 266 | 261.8 | 7.04 | 177.0 | 199.5 |
| TVC (March 15 -23, 2018) | 264 | 261.8 | 4.21 | 176.6 | 197.6 |


## 2.4 Gaussian Processes

Gaussian Processes (GP) are a non-parametric Bayesian method used in regression models. These processes are effective and flexible tools to fit complex functions with small training datasets (Quiñonero-Candela and Rasmussen, 2005). Gaussian processes provide uncertainties on predictions, using training data and prior distributions to produce posterior distributions for
predictions. Mean ($m(x)$) and covariance ($k(x, x')$) functions from the multi-variate Gaussian distribution are used to fit data (x: snow depth, y: ratio of layers). The $m(x)$ function describes the expected value of the distribution and the $k(x, x')$ describes the shape of the correlation between data points ($x_i$). Different mean and covariance kernels can be chosen to fit the data. From Bayes rule in Eq. (1) where y (ratio of layer) and X (snow depth) are observed data and *f* the GP function, posterior predictions of ratios of layers can be produced. Posterior predictions were calculated using the standard method of Markov
Chain Monte Carlo (MCMC) sampling using PyMC3 (Salvatier et al., 2016).





$$Posterior = \frac{Likelihood \cdot Prior}{Marginal\ likelihood} \quad = \quad p(f|y,X) = \frac{p(y|X,f) \cdot p(f)}{p(y|X)} \tag{1}$$

$$f(x) \sim GP\left(m(x), k(x,x'), \phi(x)\right) \tag{2}$$

Equation 2 defined $f$ as a function of $m(x), k(x,x')$. A mean function $m(x)$, following an inverse logic function ($\phi$) (Eq. 3), was chosen due to the close fit with observations. The covariance function $k(x,x')$ determines correlation between data points

($x_i$). This function is a classic Gaussian white noise covariance function and is defined with noise ($\sigma$) and the Kronecker delta function ($\delta_{x,x}$) (Eq. 4), to best fit the observations. By using a scaling function ($\phi$), the covariance function (uniform noise in this case) can be modified as a function of x. The scaling function used is also an inverse logic function ($\phi$) that takes the same form as Eq. (3). Finally, a deterministic transformation is applied to the prior (GP) to constrain values to a ratio (0,1). The likelihood of DHF observation is defined by a Beta distribution (0,1).

$$m(x) = \phi(x) = c + b\left[\frac{e^{a(x-x_0)}}{1+e^{a(x-x_0)}}\right] \tag{3}$$

$$k(x,x') = \sigma^2 \delta_{x,x'} \phi(x) \tag{4}$$

## 3. Results

### 3.1. Snow depth distribution

Distributions of snow depth are needed when integrating over large areas to calculate sub grid snow variability for distributed

models (Clark et al., 2011; Liston, 2004). The $\mu_{sd}$ and the $CV_{sd}$ of snow depth are used as parameters in probability density functions to estimate the shape of the log-normal and gamma distributions. To find which distribution best fits the depth observations, we tested the log-normal and gamma distributions using the Kolmogorov-Smirnov two sample test with snow depth observations (shown in blue in Figure 2). The statistical fits for each distribution are shown in Table 3. For both the log-normal and gamma distributions the null hypothesis is validated at the 5% significance level from p-value > 0.05 (i.e. the two

samples were drawn from the same distribution), which agrees with previous assessments of Arctic snow (Clark et al., 2011; Gisnas et al., 2016).




**Table 3: Kolmogorov-Smirnov (KS) test for 2 samples of probability distribution function (PDF).**

| Site | PDF | KS stats | p-value |
|------|-----|----------|---------|
| TVC | log-normal | 0.029 | 0.41 |
|     | gamma | 0.039 | 0.11 |
| CB | log-normal | 0.024 | 0.63 |
|    | gamma | 0.017 | 0.95 |

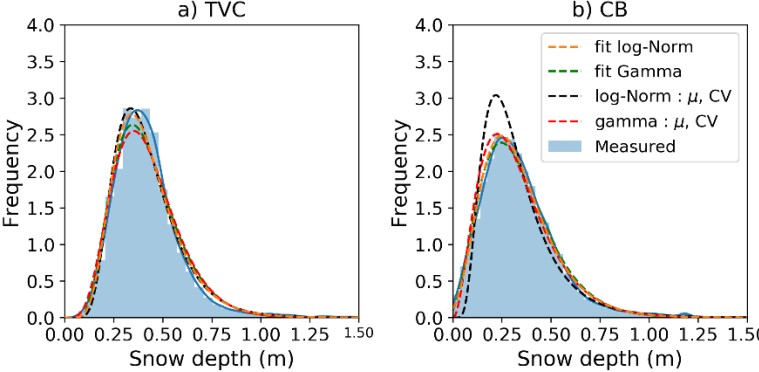

**Figure 2: Log-normal and gamma distribution fit to the measured snow depths.**


Distributions with parameterization using measured $\mu_{sd}$ and $CV_{sd}$ (Figure 2) differ from the best fit with regular parameters, especially compared with log-normal distribution in CB (black dotted line in Figure 2b). Liston (2004) reported $CV_{sd}$ of 0.4 for Arctic tundra snow, which is in close agreement with the values of 0.43 for TVC and 0.56 for CB. These values were also obtained from spatially distributed snow depth measurements around snow pits. For comparison, maps of snow depth

variability, derived using photogrammetry from a RPAS, for TVC (n = 6 325 365 with total spatial coverage of 13 km$^2$) shows a much larger $CV_{sd}$= 0.78 than magnaprobe data (n=15 731) with $CV_{sd}$ = 0.43 (Table 4). A RPAS dataset is also available for CB but with a much smaller spatial coverage (0.2 km$^2$) showing a $CV_{sd}$ of 0.49. In Figure 3, we investigated the relationship between spatial coverage of sampling and the $CV_{sd}$ parameter. Datasets include RPAS-derived data at TVC (TVC18-RPAS) containing 7 areas with various size from 1- 4 km$^2$, and at CB (map of 0.2 km$^2$), and in-situ (magnaprobe) with variable high-

density sampling over different spatial extents at Daring Lake, NWT (Derksen et al., 2009; Rees et al., 2014), Puvirnituq, QC (Derksen et al., 2010) and at Eureka, NU (Saberi et al., 2017). Results showed that the $CV_{sd}$ converges toward 0.9 as spatial coverage increased up to 4 km$^2$, suggesting typical $CV_{sd}$ values of 0.8-1.0 for microwave pixels of 3.125 km (Fig. 3).



**Table 4: Statistical parameters of snow depth distributions.**

| Site | n | $\mu$ (m) | $\sigma$ (m) | $CV_{sd}$ |
|---|---|---|---|---|
| TVC19 | 8 541 | 0.44 | 0.14 | 0.33 |
| TVC18 | 7 190 | 0.39 | 0.21 | 0.54 |
| **TVC** | **15 731** | **0.42** | **0.19** | **0.43** |
| **TVC18-RPAS** | **55 583** | **0.46** | **0.36** | **0.78** |
| CB19 | 982 | 0.42 | 0.17 | 0.40 |
| CB18 | 577 | 0.34 | 0.18 | 0.53 |
| **CB18-RPAS** | **7290** | **0.39** | **0.19** | **0.49** |
| CB17 | 4 045 | 0.42 | 0.19 | 0.46 |
| CB16 | 3 403 | 0.28 | 0.16 | 0.61 |
| CB15 | 12 282 | 0.32 | 0.18 | 0.57 |
| **CB** | **20 712** | **0.36** | **0.18** | **0.52** |


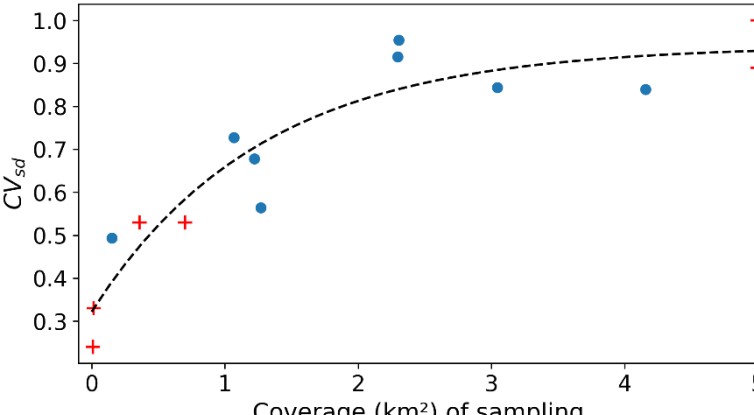

**Figure 3: Snow depth variability as a function of spatial coverage from different RPAS maps (blue round point) and in-situ sampling (red cross). The two points at the limit coverage scale correspond to areas of respectively 625 km2 ($CV_{sd}$ = 1; Daring Lake site; C. Derksen personal communication) and 198 km² ($CV_{sd}$ = 0.89, Euraka site; Saberi et al., 2007). The dotted line corresponds to the**
**exponential variogram curve to fit the data with $R^2$ = 0.88.**

### 3.2. Analysis of SSA and density per layer

After combining measurements from all snow pits at TVC and CB (n = 315) the mean proportion of DH layer thickness was 46% and WS was 54%. A small amount of surface fresh snow (SS) was present in some pits but was not included in this
calculation as this type of snow was a short-lived layer, combining fresh precipitation that rapidly transformed into rounded grains due to destructive metamorphism and defragmentation by wind. Distributions of SSA are more distinct between layers then density (Figure 4a and b), c.f. Rutter et al. (2019). Figure 4 c) and d) show that the mean values for density of WS (335 kg · m$^{-3}$) and DH (266 kg · m$^{-3}$) were closer together. SSA distributions also showed a gap between both mean values (WS:





19.7 m²kg⁻¹ and DH: 11.1 m²kg⁻¹) (Figure 4, Table 5). Even if snow properties can show high heterogeneity at local scales,
simple distributions approximate this variability well. Temporal (year) and spatial (regional between site) variation is low and
snow properties (density and SSA) can be approximated by a distribution for each distinct layer, WS and DH as in Figure 4.
Therefore, snow properties were simplified in distributions for each layer (WS and DH) representing tundra snow.

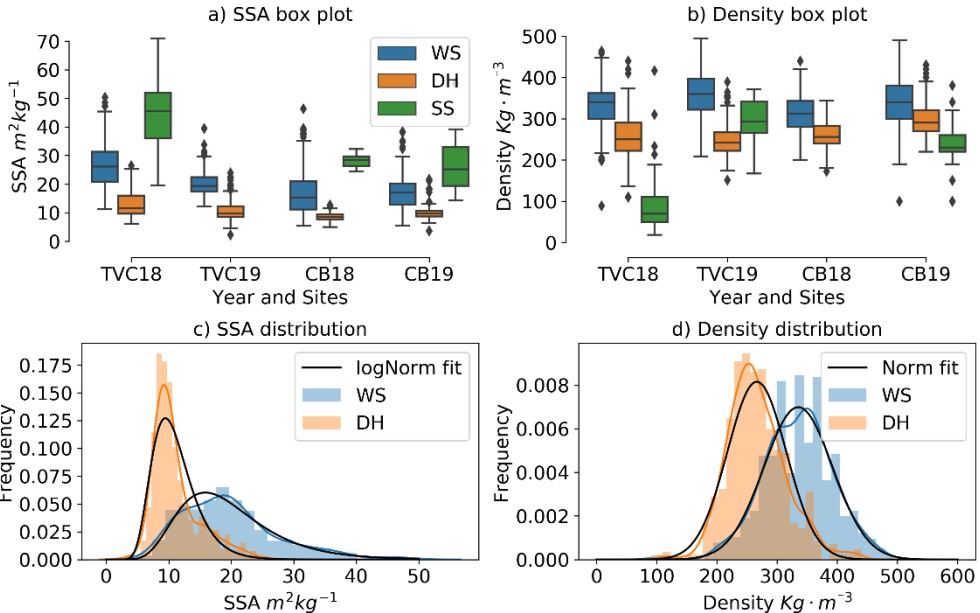

**Figure 4: SSA and density variability of Surface Snow (SS), Wind Slab (WS) and Depth Hoar (DH) for the two studied sites (TVC and CB) and different dates (see Table 5).**

**Table 5: Parameters for best fitting distribution of SSA and density for layers of DH and WS.**

| Snow property | Best fit PDF | | $\mu$ | $\sigma$ |
|---|---|---|---|---|
| **SSA ($m^2 kg^{-1}$)** | log-normal | DH | 11.1 | 3.8 |
| | | WS | 19.7 | 7.8 |
| | | | $\mu$ | $\sigma$ |
| **Density ($kg\ m^{-3}$)** | normal | DH | 266.3 | 48.9 |
| | | WS | 335.2 | 57.1 |

Layer ratios, as a proportion of total depth, showed higher variability in shallow snowpacks (Figure 5). This corroborates
measurements from Rutter et al. (2019) that suggested high variability in the DHF stabilizing at 20-30% with increasing snow
depth. Deeper snowpacks, up to 17% of the TVC basin (King et al., 2018), are a consequence of snow accumulation from


wind redistribution and compaction, explaining why deeper snowpacks (topographic drift) are dominated by wind slabs
(Benson and Sturm, 1993; Rutter et al., 2019). Also, deep snowpacks are characterized by lower temperature gradients and

consequently a reduction of kinetic growth metamorphism given the higher thermal conductivity of dense snow (Colbeck,
1983); shallower snowpacks will promote kinetic growth leading to a higher percentage of depth hoar. The mean ratio starts
around 50% (< 20cm) and reduces to 20% as snow depth increases (>1m).

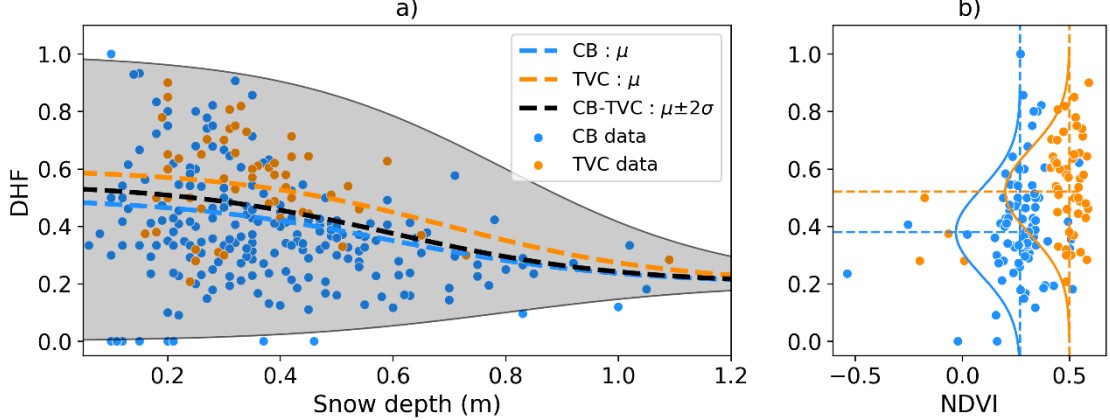

**Figure 5: Depth hoar fraction (DHF) as a function of total depth for snow pit data from 2015-2019 in Cambridge Bay and 2018-**
**2019 for Trail Valley Creek. Both datasets were separated in equal bins (10 cm) to estimate the mean value shown with dashed**
**line. The black line represents the mean for both site with the 95% interval.**

Vegetation also strongly influenced variability of DHF in shallower snowpacks, where arctic shrubs promote depth hoar
formation (Domine et al., 2016; Royer et al., 2021; Sturm et al., 2001). However, there is no clear link between DH ratios and

NDVI (a proxy for vegetation type) at local scales (Figure 5b). Since shrubs provide shelter to snow up to their own height
(Gouttevin et al., 2018), vegetation height rather than type would be required. However, at the regional scale differences are
evident between both regions, where mean NDVI and DH ratio are greater at TVC (NDVI = 0.5, DHF = 0.54) than CB (NDVI
= 0.27, DHF = 0.38).

**3.3.  DHF predictions using snow depth with Gaussian Processes**

The impact on microwave scattering of variability of layer microstructures with snow depth was previously accounted for in
Saberi et al. (2020) by defining two categories, a high scattering thin snow layer (high DHF) and a thicker self-emitting layer
(low DHF). Snowpack properties (layer extent, density, SSA) were related to snow depth via DHF (Figure 5) instead of using
two categories. Using Gaussian Processes (GP), DHF were fitted and predicted based on snow depth values (Figure 6). In

order to use GP, the mean function $m(x)$, following an inverse logic function ($\phi_1$: Eq. 3), was chosen with parameters: a = -





5, $x_0 = 0.6$, b = 0.35 and c = 0.2 to best match the mean line observation for both sites in Figure 5. The mean function set the mean value across the snow depth range. The correlation function was set to a uniform noise, but this noise was reduced from depth > 40 cm by using a scaling function ($\phi_2$: a=-5, $x_0 = 0.6$, b = 1.5 and c = 0.25). An inverse logic function ($\phi_1, \phi_2$) was used twice in the fitting 1) for the mean value and 2) to reduce the variability (noise) as snow depth increased. The snow pit

dataset (n=315, Figure 5) was used to build posterior predictions using MCMC sampling.

For prediction of DHF, any number of snow depths can feed into the posterior prediction or GP fit. Snow depths were generated from a log-normal distribution with parameters ($\mu_{sd}$, $CV_{sd}$) from previous section in Table 4. Posterior predictions of DHF were similar to observed data (Figure 6) and followed closely posterior probability representation in red (GP fit). Again, higher variability in DHF was reproduced for depths < 0.5 m, which was then reduced for depths > 0.5m following the red posterior

prediction representation in Figure 6.

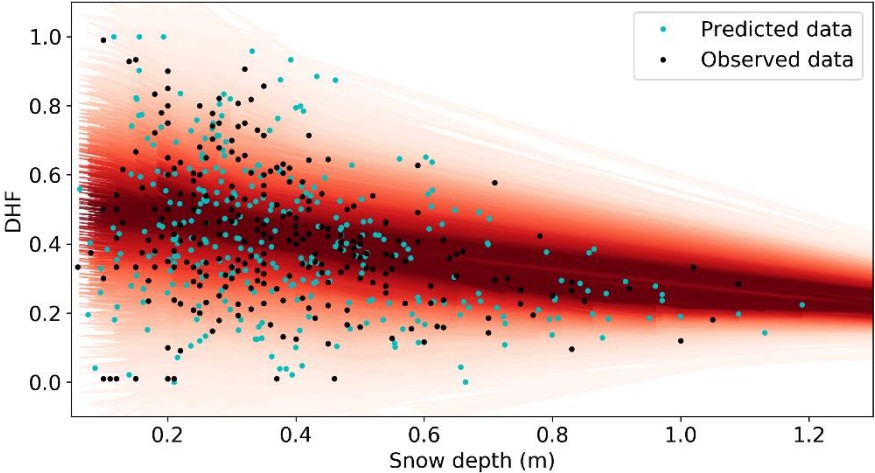

**Figure 6: Prediction on DHF (cyan) using a GP fit trained on observed data (black). Snow depth were samples from a log-normal distribution with parameters from Table 4. The GP fit is illustrated in red where darker red represents high posterior probability**
**that follows the mean function.**

### 3.4. SMRT simulation of sub-grid variability within sensor footprint

SMRT simulations using measured snowpack properties were compared with the satellite measurements of $T_B$. Two simulations were evaluated using: 1) mean measured depth, each layer's density and SSA, and DHF, and 2) a log-normal distribution of snow depth and the GP fit (predicted DHF). We hypothesized that the 3.125 x 3.125 km EASE 2.0 grid pixel

for 37 GHz can be separated into $n$ smaller sub-grid pixels. Sub-grid pixels ($n = 500$) represent the observed snow variability, where $n$ snow depths will follow a log-normal distribution with parameters $\mu_{sd}$ and $CV_{sd}$. The ratio of each layer is predicted




using the GP fit with depth as input from the log-normal distribution. Mean SSA (DH: 11 m²kg⁻¹, WS: 20 m²kg⁻¹) and density (DH: 266 kg m⁻³, WS: 335 kg m⁻³) per layer were determined from measurements (Figure 4).

For one standard EASE-grid pixel, a distribution of sub-grid $T_B$ were simulated to reproduce a realistic distribution of $T_B$ within the radiometer footprint. This variability was derived from spatially distributed observations from snow pits and snow depths observation. Snow depths followed a log-normal distribution with the mean measured depth ($\mu_{sd}$) of each region (Table 4) and a depth variability ($CV_{sd}$) that was evaluated from a range of 0.1 to 1. The GP mean function from Figure 5 was used to predict the DHF for each region. When using $CV_{sd} = 0.7$, the simulated distribution showed a wide sub-pixel variability ($\pm$

40K) with a mean value of T_B(H) = 175.5 K (red line in Figure 7a), very close to the satellite-measured T_B(H) of 176.6 K (green dotted line in Figure 7a). In this case, the T_B value simulated from the mean measured snow depth and mean DHF was slightly lower (171.5 K, i.e., a bias of 5.1 K, Table 6) (black dotted line in Figure 7a). Because the simulated T_B distribution was not exactly a normal distribution, it appeared that the mean $T_B$ of this distribution increased when $CV_{sd}$ increased (Figure 7b). This meant that snow depth variability ($CV_{sd}$) must be accounted for when estimating the average $T_B$, in addition to the

mean snow depth values. The influence of the GP simulation on the mean simulated $T_B$ was approximately 10 K (Figure 7b) as $CV_{sd}$ varies from 0.1 to 1.

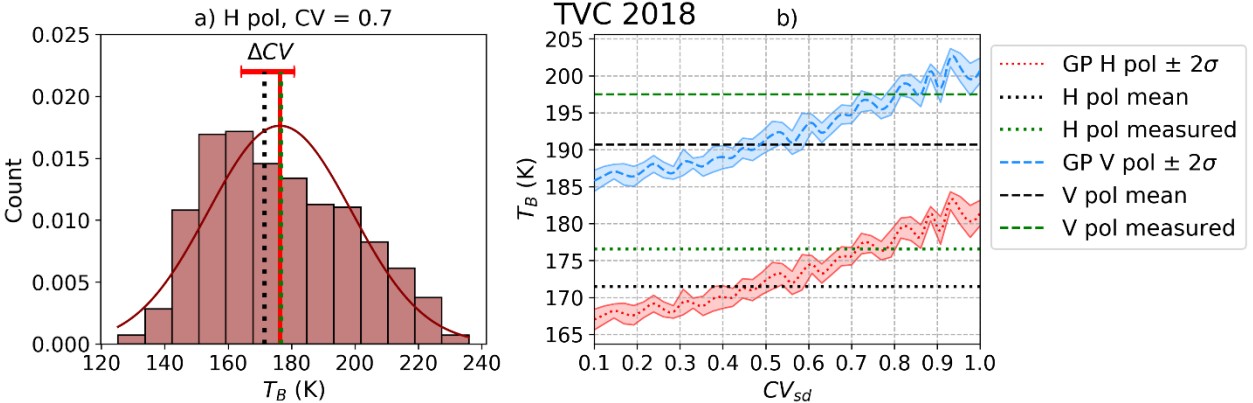

**Figure 7:** Brightness temperature variability simulation a) distribution of simulated Tb within a pixel, where vertical lines represent
the mean of this distribution for H pol (red), measured by satellite (green) and T_B value simulated from the mean measured snow depth and mean DHF (black). In b), the mean of the simulated Tb for H pol (red) and V pol (blue) as a function of $CV_{sd}$ with mean values (dotted black lines). The $CV_{sd}$ that minimized biases is located at the red/blue-green intersection. Shaded blue and red areas correspond to a $2\sigma$ range ($\pm$ 1K) representing uncertainty inherent from our Bayesian simulations in estimating the mean of simulated $T_B$ for the pixel.


GP simulation reduced biases by 5K with a higher optimized $CV_{sd}$ (intersection of red/blue - green line, Figure 7b). A similar pattern was observed for CB (not shown here) but the measured $T_B$ at CB was much higher than the GP simulation resulting





in large bias for CB (~20K) compared to TVC (Table 6). Both sites suggested a larger 0.9- $CV_{sd}$, which agreed with a $CV_{sd}$ of 0.9 for larger spatial coverage measured in Figure 3. Observed large biases at CB vary over the years from 5K to 29K. The

total RMSE of both sites and years linearly decreased as a function of $CV_{sd}$ (Figure 8). Total RMSE is minimized with higher $CV_{sd}$ (0.8-0.9) typical of large sampling scale (over 4 km$^2$) as shown in Figure 3.

**Table 6: Bias between SMRT simulated and measured Tb from SSMIS sensor at each site.**

| | | Bias (K) | | | | RMSE (K) | |
|---|---|---|---|---|---|---|---|
| | | CB | | TVC | | | |
| SMRT simulation type | Year | H pol | V pol | H pol | V pol | H pol | V pol |
| | 2019 | 28.2 | 25.9 | 6.9 | 10.3 | 17.8 | 19.1 |
| | 2018 | 8.0 | 5.3 | 5.1 | 6.8 | | |
| mean depth and DHF | 2017 | 19.9 | 18.9 | - | - | | |
| | 2016 | 16.9 | 23.2 | - | - | | |
| | 2015 | 24.7 | 29.1 | - | - | | |
| | 2019 | 18.6 | 15.7 | -4.4 | -1.2 | 9.7 | 10.4 |
| | 2018 | -3.7 | -6.2 | -4.9 | -3.2 | | |
| GP simulation CV = 0.9 | 2017 | 10.4 | 9.3 | - | - | | |
| | 2016 | 7.1 | 13.5 | - | - | | |
| | 2015 | 10.0 | 13.9 | - | - | | |


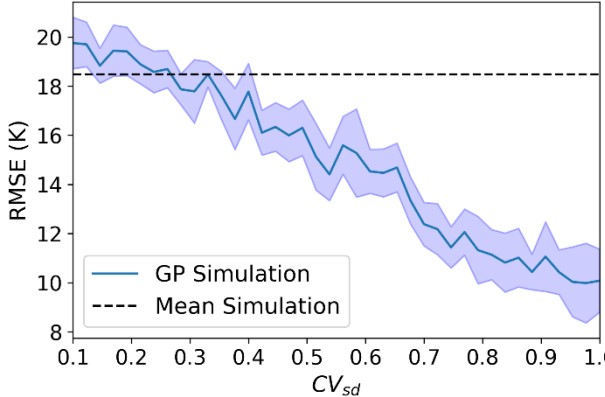

**Figure 8: Overall RMSE (year and site) with the mean simulation and the GP simulation in blue as a function of the coefficient of variation.**





## 4.   Discussion

A strong link between spatial coverage and $CV_{sd}$ was found. As spatial coverage increased to 3-4 km², the $CV_{sd}$ parameter also increased to values of ~0.9 which matches the optimized values found in this study using GP simulation. This indicates that a $CV_{sd}$ between 0.8-0.9 is desirable to represent snow depth variability in SWE retrievals since the pixel size of passive microwave products are 3.125 km for the EASE GRID 2.0 at 37 GHz. However, the resolution of SWE products like GlobSnow 3.0 are much larger (25km); future investigation of $CV_{sd}$ values at those scales have the potential to help GlobSnow 3.0

(Pulliainen et al., 2020). For small areas (< 0.5 km²), $CV_{sd}$ values were low around 0.3-0.4 and similar to the mean simulation (Figure 8). This indicates that at scales of < 0.5 km², the mean of snow depth is sufficient to represent snow in SMRT (passive mode), but as the spatial coverage increases variability also increases and suggests a second parameter ($CV_{sd}$) is needed to represent snow variability. The link between spatial coverage and snow depth variability can be used to improve land data assimilation (Kim et al., 2021) depending on the scale used for the application.


Spatial complexities of Arctic snowpacks can be adequately characterized with distributions of snow depth (Figure 2) and simplified by considering density and SSA of two main layers (Figure 4). Such simplifications could be potentially useful for satellite SWE retrievals across Arctic tundra regions. Since Bayesian SWE optimization needs a strong first guess from regional *a priori* information, multiple distributions of snow depth, density and SSA presented here can be used for tundra

type snow in MCMC sampling (Pan et al., 2017; Saberi et al., 2020). Additionally, a similar approach to our GP simulation can be added so the $CV_{sd}$ parameter can also be used as *a priori* information with a distribution from 0.8 to 1, since it improved T$_B$ RMSE by ~8K (Figure 8). This approach improved $T_B$ simulation compared to using only mean values of snowpack properties by adding variability within the footprint. The $CV_{sd}$ parameter (describing variation in snow depth) has a considerable effect on brightness temperature (10 K) when used as an effective parameter to account for sub-pixel variability

of snow depth. The amount of scatterers (snow grain and structure) within the radiometer's footprint is adjusted via the DHF predicted from snow depth ($CV_{sd}$). This idea of modulating the amount of scatterers based of DHF prediction from a GP and a distribution of snow depth ($\mu_{sd}$ and $CV_{sd}$) can be extended to future active Ku-band mission (Garnaud et al., 2019; King et al., 2018) as it known that microwave spatial variability affects backscatter signal (King et al., 2015) and SWE retrievals (Vander Jagt et al., 2013). The $CV_{sd}$ parameter is proposed as an effective parameter to account for variability inside the grid

cell, while the mean depth ($\mu_{sd}$) is dependent on precipitation at a larger scale, in situ measurements at weather stations in data assimilation schemes (Takala et al., 2011), or by physical snow model (Larue et al., 2018). The $CV_{sd}$ could be set using relations with spatial coverage (Figure 3) or estimated from statistical topographic relation (Grünewald et al., 2013).



## 4. Conclusion

This study evaluated the use of parameters controlling snow depth distributions to improve passive microwave SWE retrievals
by characterizing tundra snow sub-pixel variability. In shrub and graminoid tundra environments, mean values of snow depths
($\mu_{sd}$ = 0.33-0.44m) and coefficient of variations ($CV_{sd}$ = 0.4-0.6) were similar to those previously reported in Arctic tundra
(Derksen et al., 2014; Liston, 2004; Sturm et al., 2008). However, a substantial difference in TVC-2018 between point
observations of snow depth (magnaprobe; $CV_{sd}$ = 0.54) and the spatial maps of snow depth (RPAS; $CV_{sd}$ = 0.78) indicates a
potential underestimation of the $CV_{sd}$ parameter. An increase in $CV_{sd}$ matches increase in spatial coverage of snow depth
sampling, indicating that a higher $CV_{sd}$ (0.9) is more suited to estimate snow depth variation in the 3.125 km resolution EASE-
Grid 2.0. The $CV_{sd}$ was shown to be an effective parameter to account for snow depth variability in simulation of snow $T_B$. A
two-layer snowpack model (depth hoar and wind slab), which contains snowpack properties simplified into distributions, was
used to initialize the SMRT model via a GP fit of the DHF related to snow depth. DHF is fitted to snow depth using a Bayesian
Gaussian Process, which accounts for variation in snow scattering using $CV_{sd}$. The parametrization of the Improved Born
Approximation ($\kappa_{37}$ = 1.39) microstructure model and grain size (Vargel et al. 2020) was used successfully to simulate satellite
$T_B$, but there is still substantial uncertainties in the simulated values which are likely to be linked to microstructural properties
not captured by SSA (Krol and Löwe, 2016). SMRT simulations of $T_B$ were reduced by 8 K after optimizing $CV_{sd}$ to higher
values (0.8-1.0), thereby matching $CV_{sd}$ of spatially distributed snow depth from RPAS, and acting as an effective parameter
to compensate for variation in snow properties inside the footprint of satellite sensor. The $CV_{sd}$ parameter is proposed as an
effective parameter to account for variability inside the footprint to minimize the difference between microwave measurements
and simulations in SWE retrievals algorithm. Difference minimization would be beneficial to the data assimilation scheme of
the European Space Agency: GlobSnow product (Takala et al., 2011) and modelled large scale climate trend products
(Mortimer et al., 2020; Pulliainen et al., 2020) of tundra snow.

*Data availability*

Data and code for the gaussian process fit and GP simulation are available on
https://github.com/JulienMeloche/Gaussian_process_smrt_simulation. RPAS map and magnaprobe from TVC are available at
https://doi.org/10.5683/SP2/PWSKKG.

*Author contributions*

JM: Formal analysis, Investigation and writing - original draft preparation, AL: Writing – review & editing, Supervision,
Investigation, Funding acquisition and Resources, NR: Writing – review & editing, Supervision, Investigation, AR: Writing –
review & editing, Supervision, Investigation, JK: Writing – review & editing, Data acquisition, BW: Writing – review &
editing, Data acquisition.



*Acknowledgements*

The authors would like to thank the entire GRIMP research team from Université de Sherbrooke and CHARS staff team for field work assistance from 2015-2019 in Cambridge Bay. We thank Phil Marsh (Wilfrid Laurier University), Chris Derksen and Peter Toose (Environment and Climate Change Canada) for logistics and field work at Trail Valley Creek Research Station. We also thank Victoria Dutch from Northumbria University for her help with dataset management of TVC. This research was made possible thanks to the financial support of the Natural Sciences and Engineering Research Council of Canada (NSERC), Polar Knowledge Canada, the Canadian Foundation for Innovation (CFI), Environment and Climate Change Canada (ECCC), Fonds de recherche du Québec – Nature et technologies (FRQNT), Northern Scientific Training Program (NSTP) and research funding from Northumbria University, UK.

*Competing Interests*

The authors declare that they have no conflict of interest.

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
