# Peer review of "Characterizing Tundra snow sub-pixel variability to improve brightness temperature estimation in satellite SWE retrievals"

_The Cryosphere, 2021_

## Referee Comment (RC1)

Review of Characterizing Tundra snow sub-pixel variability to improve

brightness temperature estimation in satellite SWE retrievals. (tc-2021-156)

I found this to be an interesting and reasonably well-written paper. I will restrict my comments to primarily the nature of tundra snow, and the remote sensing of tundra snow, as I am not sufficiently well-versed in Bayesian statistics to comment on those aspects of the paper.

Overall, the paper is a useful addition to microwave remote sensing as well as providing characteristics of tundra snow. The two basic findings are that a) the coefficient of variation (CV) can be used to introduce sub-grid (<3 km) scale realism into microwave retrieval models, and b) using some form of a snow depth-to-depth hoar relationship, the extreme microwave scattering of the depth hoar (and implicitly, the self-emission of the wind slab) can be approximated when using these models. The authors claim an improvement in SWE retrievals (Tb) by (8K). This finding would have been more useful if expressed in terms of depth or SWE improvement.

What is missing from the paper is some deeper inductive reasoning that could take the work farther and make it more general (and less about two particular tundra locations). Personally, I found Figure 3 the most interesting result in the paper and found myself wondering why the CV (as a function of the area measured) appeared to be asymptotic to 1. It was not that I doubted the data, but I wondered if that was some physical limitation to CV, or just some limitations in the available data. The authors stated in their discussion that:

However, the resolution of SWE products like GlobSnow are much larger (25km); future investigation of CVsd values at those scales have the potential to help GlobSnow 3 (Pulliainen et al., 2020).

and I agree with this statement, but suggest we hardly need to wait for future investigations. I would suggest the authors could address this issue more thoughtfully in this paper using the knowledge base they already have. Let's start (Table below) by examining some extreme depth distributions using Excel. For a completely homogeneous snow depth field, the CV approaches zero. For more realistic heterogeneous snow, and certainly most tundra snow fields, the CV rises with area because (I believe) of snow drifts. For example, in a landscape of mostly very thin snow with with a few very deep drifts I was able to produce values >4 (Case 7). This is exactly the type of situation that exists in tundra snow, particularly in the windier tundra areas (e.g. the Arctic Refuge in Alaska and in the Barrenlands of Canada) where wind scour and drifting is most extreme. I suspect CV values over 2 are often realized, for example the tundra landscape shown below (after the thin snow has melted):

---

## Author Comment (AC1)

**Characterizing Tundra snow sub-pixel variability to improve brightness temperature estimation in satellite SWE retrievals**

Manuscript ID: tc-2021-156

Note to editors:

The following co-authors will be added to the manuscript: Philip Marsh and Evan Wilcox.

We thank all reviewers for the helpful and constructive comments that helped improve the manuscript. A new figure was added to the manuscript introducing a new dataset in TVC with addition of Monte Carlo simulations involving $CV_{sd}$ and $\mu_{sd}$. The 19 GHz frequency was also added in the GP simulation of Figure 7. See revised manuscript for updated numbering of figures. Every comment was addressed, and a detailed list of modifications is provided below.

Reviewer's comments

Answers to reviewer

*Addition to text or* original text with *added text*

**Reviewer: 1**

R1.C1 What is missing from the paper is some deeper inductive reasoning that could take the work farther and make it more general (and less about two particular tundra locations). Personally, I found Figure 3 the most interesting result in the paper and found myself wondering why the CV (as a function of the area measured) appeared to be asymptotic to 1. It was not that I doubted the data, but I wondered if that was some physical limitation to CV, or just some limitations in the available data. The authors stated in their discussion that: *However, the resolution of SWE products like GlobSnow are much larger (25km); future investigation of CVsd values at those scales have the potential to help GlobSnow 3 (Pulliainen et al., 2020).* and I agree with this statement, but suggest we hardly need to wait for future investigations. I would suggest the authors could address this issue more thoughtfully in this paper using the knowledge base they already have. Let's start (Table below) by examining some extreme depth distributions using Excel. For a completely homogeneous snow depth field, the CV approaches zero. For more realistic heterogeneous snow, and certainly most tundra snow fields, the CV rises with area because (I believe) of snow drifts. For example, in a landscape of mostly very thin snow with with a few very deep drifts I was able to produce values >4 (Case 7). This is exactly the type of situation that exists in tundra snow, particularly in the windier tundra areas (e.g. the Arctic Refuge in Alaska and in the Barrenlands of Canada) where wind scour and drifting is most extreme. I suspect CV values over 2 are often

realized, for example the tundra landscape shown below (after the thin snow has melted):But the authors need not just deal with this CV issue in a theoretical framework: they should have access to the TVC lidar maps we produced in 2012. They could readily run a Monte Carlo simulation, varying the location and area examined, then plot the resultant mean depths and CVs thereby adding to the figure. Once that was done, they could move to more general application of CV to the full range of tundra snow. By the way, a quick look at Wikipedia indicates that for small samples, CV is low-biased.

We agree this part of the paper is interesting and could be improved by Monte Carlo simulations using the Lidar dataset presented in Rutter et al. (2019). We followed the recommendations suggested and did Monte Carlo simulations varying the location and area of sampling using uniform randomly generated radius and location of a circle (mask) using the Lidar dataset. We also aggregated the multiple maps from TVC in 2018 (Walker et al., 2020) to perform the same analysis. Both the mean and CV were evaluated and are shown in the updated figure 3. Multiple addition in the text were done.

First, the data section was modified to add the dataset TVC13-Lidar. See modified Table 1 and Table 4 in the revised manuscript for addition of TVC13-Lidar

*A Lidar dataset of TVC snow depths (93 km$^2$ at 10 m resolution) from April 2013 (Rutter et al., 2019) was also used. Monte Carlo simulations of both the $\mu_{sd}$ and $CV_{sd}$ were performed on each snow depth map. Simulations randomly selected pixels as the center of a circular mask with a random radius. The mask was used to select all pixels within the circle so the statistical parameters ($\mu_{sd}$ and $CV_{sd}$) could be calculated.*

This text was added in section 3.1 along with the new figure 3. Previous figure was removed.

*[…] the larger lidar derived snow map from TVC in 2013 was used. Figure 3a) shows snow accumulation of TVC13-Lidar and TVC18-RPAS with snow drift visible in dark blue and Sub-grid of 1km$^2$ showed areas with high $CV_{sd}$ (Figure 3b) containing more drift. For both areas, 500 Monte Carlo simulations were performed by randomly selecting sub-regions within each domain (Figure 4) so the mean and variability as a function of coverage could be investigated. Simulations showed sub-sampling of $\mu_{sd}$ and $CV_{sd}$ converged to the values of the full area. The mean of each area was similar in value with less variation in the simulations compared to $CV_{sd}$. A difference of 0.2 between the full $CV_{sd}$ of the RPAS (5 km$^2$) and Lidar (93 km$^2$) maps (Figure 4) was found.*

[Figure]

*Figure 1: RPAS and Lidar dataset of snow depth at TVC (TVC13-Lidar and TVC18-RPAS). TVC13-Lidar is the largest dataset covering 93 km². TVC18-RPAS is a smaller dataset within the area of TVC13-Lidar. In a) is shown the snow depth map at 10 m resolution from 2013. b) and c) show a sub grid of 1 km with $CV_{sd}$ and $\mu_{sd}$ within each cell.*

[Figure]

*Figure 4: Snow depth mean ($\mu_{sd}$) and variability ($CV_{sd}$) as a function of coverage for sampling area. Monte Carlo simulations were done using the two datasets in TVC from Figure 3, TVC13-Lidar and TVC18-RPAS. The multiple maps from TVC18-RPAS, CB18-RPAS and in-situ sampling from other studies were also added (yellow square and red cross). The $\mu_{sd}$ and $CV_{sd}$ of both full areas are shown by the black dotted and dashed line.*

We initially thought the CV would increase as spatial coverage increased. Instead, the lack of data points hid high variability in CV for small areas found from the Monte Carlo simulation of both dataset (TVC18-RPAS and TVC13-Lidar). As mentioned by the reviewer, the CV values depends on whether there is enough drift capture in the area sampled. The following was added in the discussion to address Figure 4.

*As spatial coverage increased, the $CV_{sd}$ parameter converged to the full area values (Figure 4). Simulations showed high variation in $CV_{sd}$ (from 0.1 to 2) for areas < 10 km². Snow accumulation varied at the meso scale (100 m to 10 km) due to topography and vegetation (Pomeroy et al., 2002) by varying wind-flow direction (Liston and Sturm, 1998). At the meso scale, variability in $CV_{sd}$ was high due to topographic differences; plateau, slope and valley create favorable conditions from wind flow direction to promote snowdrift, scour and sublimation processes (Parr et al., 2020; Rutter et al., 2019). Vegetation facilitates snow holding capacities by decreasing wind speed near*

*the ground within and downwind of shrub (Marsh et al., 2010; Sturm et al., 2001). Some areas include both extreme drifts and thin snow , resulting in high $CV_{sd}$ (dark green areas in Figure 3b) which are commonly found in TVC (Walker et al., 2020). $CV_{sd}$ was lower for areas without drifts (light green areas in Figure 3b). In areas $> 10\ km^2$ (Figure 4d), variation in $CV_{sd}$ is reduced and yielded higher values.*

Also, the following paragraph in the discussion was completely removed and modified as follow.

*Convergence to higher $CV_{sd}$ as spatial coverage increased matched the PMW optimized values found in this study using GP simulation (0.8 – 1.0). Our analysis in Figure  d) showed that $CV_{sd}$ of TVC13-Lidar converged to 0.6 at 93 $km^2$, but two in situ points from other studies at 625 $km^2$ had higher $CV_{sd}$ (0.9-1). This indicates that a $CV_{sd}$ between 0.6-1.0 is desirable to represent snow depth variability in SWE retrievals for PMW SWE products at 25 km for the EASE GRID 2.0 and 625 $km^2$ for GlobSnow 3.0 (Pulliainen et al., 2020) . For active sensors (resolution $< 1$ km), the high variability in $CV_{sd}$ under 1 $km^2$ due to high variation in snow depth (Figure b) can affect back scattering since active sensor at Ku band are also sensitive to volume scattering (King et al., 2018). The need for prediction of $\mu_{sd}$ and $CV_{sd}$ based on topography could become essential at these scales not only for microwave remote sensing but also snow modelling or land data assimilation (Kim et al., 2021).*
* * *
R1.C2 The other aspect of the paper that bears some thought, and is related to the above point, is how wind slab and depth hoar fractions must interact. Step 1 in approaching this would be to explain in greater detail how those types of snow were identified in the snow pits in this study. I was struck by the relatively close density values reported in the study for depth hoar and wind slab (means 266 vs. 335 kg/m 3 ). The former value is typical for mildly indurated tundra depth hoar, but the latter is quite low for tundra wind slab, which can exhibit values over to 550 kg/m 3 . Wind slabs of 300 kg m 3 are often soft and hardly wind-worked at all, and in addition, many less experienced field practitioners fail to note small and newly faceted grains in wind slab of this nature. Then there is the problem of "indurated depth" hoar (Sturm et al., 2008; Derksen et al., 2009; Domine et al. 2018), snow layers that were wind slab but have metamorphosed into depth hoar. Presumably the critical aspect of differentiating these textures for microwave remote sensing is that the ornate, hollow and plate- like depth hoar grains scatter microwaves far better than the wind slab, hence subdividing the pack into those two fractions is critical. The relatively similar values of SSA (Figure 4a) for slab and hoar suggest to me the authors were dealing with a of properties rather than a truly distinct bimodal snow pack. I went back to the paper the authors referenced related to a two-component snow model they used:

Saberi, N., Kelly, R., Toose, P., Roy, A., Derksen, C., 2017. Modeling the observed microwave emission from shallow multilayer Tundra Snow using DMRT-ML. Remote Sens. 9. https://doi.org/10.3390/rs9121327

and was pleased to see that a long-forgotten paper of mine

Sturm, Matthew, Thomas C. Grenfell, and Donald K. Perovich. "Passive microwave measurements of tundra and taiga snow covers in Alaska, USA." Annals of Glaciology 17 (1993): 125-130.

had been used in developing that model. That work showed that depth hoar volume scattering was more than 6X effective compared to windslab. It should be possible to go beyond the findings of Rutter et. al. (2019) for TVC, where the DHF was shown to stabilize at 30% for depths over 60 cm, but not why. Figure 2 in this paper shows for both study sites long tails on the distributions out to 150 cm, while the mean depth appears to be 1/3 rd of that value. In a recent paper Parr et al. (2020) defined a drift depth threshold as being approximately the mean plus 1s, so that "extra" depth is statistically likely to be transported snow. A different way to look at Figures 5 and 6 is that for the mean snow depth half the pack is depth hoar; where the pack as been scoured (drift snow removed) that fraction is higher; where the snow is drifted, that fraction is lower. Perhaps the fraction where it is lower would be the mean plus 1s... I am not sure. But some attempt to understand the processes behind the statistics (Bayesian or otherwise) could help generalize the results beyond to very specific tundra locations.

For the first part of the comment, we agree that a more detailed explanation of the DH/WS classification is necessary. We added details on the multiple layers found and how they were classified as slab and hoar. The relatively close peak of each distribution can be explained by the classification of indurated hoar as DH. Also, every layer that did not contain enough large crystals were considered WS which is more a general slab (soft to hard) rather than a wind slab with high density ($> 400 \ kg \cdot m^{-3}$). The following was added in the result section 3.2.

*The goal was to classify DH as large grained snow (large facets, depth hoar cups and chains), then all other snow layers above the DH as wind slab (WS). Some layers were more difficult to classify as they contained mixed crystals or were a transitional slab-to-hoar layer (also referred to as indurated hoar) (Sturm et al., 2008). Slab that contained small faceted crystals (< 2 mm) were classified as WS. Indurated hoar, a wind slab metamorphosed into depth hoar, was classified into DH with a typical density ~ 300 $kg \cdot m^{-3}$. Because of this reason, the peak of each distribution appeared close to each other in Figure 5 c) and d). For retrieval of snow properties using satellite remote sensing, a 2 layer radiative transfer model using WS and DH can be used to simplify much of the layer complexity found in arctic snowpacks (Rutter et al., 2019; Saberi et al., 2017).*

The second part refers to the relationship between DHF and snow depth and how we could go beyond the statistical fit by investigating the process behind the statistic by leveraging Parr et al. (2020). An attempt in understanding the process from your comments was added in section 3.2.

*Parr et al. (2020) found a key threshold of $\mu_{sd} + 1\sigma_{sd}$ to define snow drifts in tundra environments. This threshold of > 0.6 – 0.8 m, based on data presented in Table 4, is an important metric in **Error! Reference source not found.**6 since above this depth, the variability and the mean DHF is greatly reduced as the snowpack is dominated by wind slab for larger depth. As defined in Parr et al. (2020), the transported snow from wind accumulates at these particular*

*locations (drift) where it was scoured or removed from wind affected area yielding lower depth with high DHF.*
* * *
R1.C3 Figure 1: I tend to see light as high and dark as low.

See revised manuscript for modification in figure 1.
* * *
R1.C4 Line 187: The black line is dashed not dotted.

Done, see revised manuscript
* * *
R1.C5 Figure 4: The orange and blue fit lines are not defined.

The following was added to the caption of figure 4 (5 new version).

*Figure 5: SSA and density variability of Surface Snow (SS), Wind Slab (WS) and Depth Hoar (DH) for the two studied sites (TVC and CB) and different dates (see Table 5). ==In c) and d), the best fit distribution is shown in black with the kernel density estimate (KDE) of the histogram of each layer.==*
* * *
R1.C6 Figure 5b: For much of tundra snow, tussocks rather than shrubs, are a control on the DHF. Also, since shrubs can be layed down under the snow (and frequently are), a relationship between depth hoar and/or wind slab and NDVI seems tenuous at best.

Agreed that the relation with NDVI can be tenuous but we still think it can help at regional scale as a measured of vegetation (shrub and tussock). Both vegetation can favor growth of depth hoar with a high DHF (table 2, Sturm et al., 2001) where both vegetation have a high DHF. The point we were trying to make with this figure is that DHF potentially follows vegetation and latitudes at the regional scale. It matches nicely with a recently found results from figure 5 in Royer et al. (2021). The following was added in section 3.2.

*However, at the regional scale differences are evident between both regions, where mean NDVI and DHF are greater at TVC (NDVI = 0.5, DHF = 0.54) than CB (NDVI = 0.27, DHF = 0.38). ==This may add to the latitudinal gradient in Royer et al. (2021) where DHF follows a gradient along a northward transect of arctic sites in Québec and Nunavik. Sites at lower latitudes and with shrubs and tussocks, had higher DHF.==*
* * *
R1.C7 Figure 6: My ignorance...but does the Bayesian approach really improve the model much over just using the results of Figure 5?

No it probably doesn't improved simulation other than the relation found in figure 5 (old version). A classical approach could be used as well but our approach provides uncertainties for our simulations from the variability in DHF found at both sites. Also, a Bayesian gaussian process could be implemented in current SWE retrieval framework based on Bayesian framework (Takala et al., 2011). The following was added in the discussion about the method.

*The amount of scatterers (snow grain and structure) within the radiometer's footprint is adjusted via the DHF predicted from snow depth ($CV_{sd}$).* ==The relationship found in Figure 6 used to predict DHF (Figure 7) could also be used deterministically with the mean function ($\phi_1$) or a linear relation of DHF decreasing from 50% to 20%. However, the Bayesian gaussian process was used because SWE retrievals are currently implemented in a Bayesian framework (Takala et al., 2011).==
* * *
R1.C8 Figure 7: In these simulations, are the amalgamated results for the sub-grid pixels combined linearly, and if so, is that what happens in a microwave sensor? Is it possible to have the net result a non-linear combinations?

The following was added to specify our assumption about the effect of the sub-pixel within the sensor. We have no evidence to support our claim that the sub-pixel combined linearly but it is the assumption that we chose.

==*To represent the signal measured by the sensor, the mean of the simulated $T_B$ was chosen and it was assumed that the sub-pixels effect combined linearly at this scale in the sensor.*== *Because the simulated $T_{B37V}$ distribution was not exactly a normal distribution, it appeared that the mean $T_B$ of this distribution increased when $CV_{sd}$ increased (Figure 8b)*
* * *
R1.C9 Line 335: "... while the mean depth (sd) is dependent on precipitation at a larger scale...". This is categorically NOT true for much tundra snow, wear I would contend that wind plays as strong, and sometimes stronger, role than the mean precipitation within a domain.

This statement was removed from the sentence and now reads as follow.

*while the mean depth ($\mu_{sd}$) is assimilated by in situ measurements at weather stations in data assimilation schemes*
* * *
R1.C10 Line 344: "...potential underestimation of the CVsd parameter." See above discussion of CV. The issue of what constitutes a representative domain (or snow landscape) is thorny. Clearly if a domain fails to include, say drifts, the CV will be too low. Likewise, if the domain is limited to a coupled drift and scour zone it will be too high.

See addition from R1.C1

Also, this part of the conclusion was modified as follow.

*Monte Carlo simulations were applied to investigate $\mu_{sd}$ and $CV_{sd}$ as a function of spatial coverage. An increase in $CV_{sd}$ matched increased spatial coverage of snow depth sampling, indicating that a higher $CV_{sd}$ (0.6-0.9) is more suited to estimate snow depth variation in the 3.125 km resolution EASE-Grid 2.0. Also, simulations showed high variation in $CV_{sd}$ (> 0.9) for areas < 10 km² indicating a need for topography-based prediction of $\mu_{sd}$ and $CV_{sd}$ at this scale.*
* * *
**Reviewer: 2**

R2.C1 Line 120, could you provide more details in the ocean/lake effect removal? Although the SSMIS observations has been downscaled to 3.125 km resolution, however considering the bigger footprint of 36.5 GHz (4*6 km^2), can the water effect truly be excluded in the pixels near the ocean/lake? As can be seen from Figure 1, at CB for example, there are truly only a few grids that are lake free. How the influence of lake was considered?

Agreed, the influence of ocean or liquid water from deep lakes cannot be excluded if the pixel is 25 km (full resolution) from the coast. The snowpit area is within 25 km from the ocean so another area just outside 25 km was chosen. We skipped over this explanation originally. Our goal was to select a typical "arctic snow" area with so we could evaluate the $CV_{sd}$ and DHF effect in PMW simulation. The following was added in section 2.3 to clarify water contribution.

*For CB, an area with the same spatial coverage but a slightly different location was used since the snowpit area was within 25 km (resampled pixel resolution of SSMIS) from the ocean. The lakes in CB shown in **Error! Reference source not found.** were not considered in the soil emission contribution because most of the water was frozen (4-6) (Mironov et al., 2010), which had a similar permittivity to frozen soil (2-4) (Mavrovic et al., 2021) than liquid water.*

*However, this simplification had importance for 19 GHz given that soil emission has a greater influence on the signal at this frequency, hence the composition of frozen water and soil derived from landcover information should be used instead. Since 37 GHz is more sensitive to snow volume scattering, this step was neglected. The 19 GHz frequency was briefly used in this study in **Error! Reference source not found.** only for TVC in 2018 to investigate the effect of snow variability which modifies the amount of snow scatterers inside the radiometer's footprint.*
* * *
R2.C2 Line 120, also, the snowpit measurements were at point scale whereas the Tb data is at 3.1.25 km. Why and how the Tb data was averaged to match the point scale measurements? To which resolution was it averaged?

The following was added in section 2.3.

*for both TVC and CB regions. A single value of measured $T_B$ (per frequency) were used by averaging all pixels within snow pits area (CB: 24 pixels, TVC: 14 pixels for 37 GHz).* *Each pixel with at least one snow pit inside was used. Since all snow pits were aggregated to obtain mean value and distribution of snow properties for SMRT, averaged $T_B$ covering the snow pits area was used.*
* * *
R2.C3 Line 249: this line reads like the density and SSA of each of the two layers were estimated as a function of snow depth and DHF, too.

This sentence was removed for clarity.
* * *
R2.C4 To my understand, the DHF was determined only by one parameter, i.e., the snow depth. The prior information is the probability distribution of snow depth and the relationship between DHF and snow depth described in Figure 5. Therefore, the generated DHF (posterior DHF field) described in Figure 6 has also some random characteristics. In other words, Figure 6 is only a realization of DHF, one of the possibilities. The scatter points are not fixed, determined values. Therefore, will a different realization influence your TB simulation results?

Your understanding is correct. Different realizations are shown by the $\pm 2\sigma$ region in Figure 8 and 9. It is not explained in the text but the uncertainty ($2\sigma$) is estimated by generating the same experiment of simulating $T_B$ for the $CV_{sd}$ range of 0.1 to 1 (basically Figure 7b) 20 times. The mean and std of those 20 simulations are shown by the middle line and the $2\sigma$ range of those realizations.
* * *
R2.C5 Figure 7, it will be more interesting to provide an estimation of distribution of TB difference between 18.7 and 36.5 GHz. The authors need to explain why the TBthat considers the sub-pixel variability is higher when the standard deviation of snow depth is higher. Is it because when the snow depth is higher, the reduced variability of DHF will result in less samples of strong volume scattering, such that the TB at 36.5 GHz will increase? In addition, will this result be influenced by the soil emission background?

We decided to briefly add 19 GHz in figure 7 (old version) so the small effect (negligeable) from CV on 19 GHz simulation could be shown. See addition from comment R2.C1 about soil contribution and the addition of 19 GHz in the data section. Figure 7 was updated so 37 and 19 GHz are both shown for TVC18.

[Figure]

This part was added to the result section 3.4.

*The addition of snow variability in simulation (**Error! Reference source not found.** c-d) of 19 GHz has negligeable effect on $T_{B19}$ and showed a constant simulation across the $CV_{sd}$ range of 0.1 to 1. Simulation of $T_{B19}$ showed higher biases at horizontal polarization then vertical polarization.*

To address the second questions in the comment, the following paragraph was added in the discussion (section 4)

*Considering that the difference between 19 and 37 GHz is used in SWE retrievals (Takala et al., 2011), using the $CV_{sd}$ to account for variability of scatterers only affected simulation of 37 GHz with no effect on 19 GHz (**Error! Reference source not found.**). If standard deviation of snow increases (more drift) then relatively fewer large scatterers from depth hoar are present within the footprint due to a low DHF in large drifts. The net result is then an increase in $T_B$ at 37 GHz resulting from an increase in $CV_{sd}$ (**Error! Reference source not found.**).*
* * *
R2.C6 How the effect of vegetation was considered in the simulation?

The effect of vegetation was not considered because it is not accounted in tundra snow retrievals (Saberi et al., 2020). Shrubs and tussock are not considered as trees or tall vegetation with significant interaction. Some studies do account for vegetation interaction with PMW but in subarctic areas with trees (Derksen et al., 2012; Larue et al., 2018; Roy et al., 2012). The interaction is based on vegetation product like Leaf Area Index which are not available for small vegetation like shrub.
* * *
R2.C7 Line 25: Snow depth simulations ---> do you mean the retrieved snow depth, or the brightness temperature simulations?

This sentence was modified.

*==SMRT== simulations using a $CV_{sd}$ of 0.9 best matched $CV_{sd}$ observations [...]*
* * *
R2.C8 Line 40: dielectric properties ---> suggested to change to radiometric properties

Modification done.
* * *
R2.C9 Line 75: More words is need to explain the Gaussian Process (GP) when this term first appears here. Maybe it is better to first mention it between lines 60-65.

The following sentence was modified by removing Gaussian Processes to avoid confusion.

*which suggested the use of a term ==involving variation in snow depth and microstructure within the footprint== instead of a uniform snow depth.*

This sentence was also modified in the next introduction paragraph when stating the objectives of the study.

*Finally, we perform a Gaussian Process fit to estimate depth hoar fraction (DHF) from snow depth, using probability density functions of snow depth ==to add variation of snow depth and microstructure within the footprint==.*
* * *
R2.C10 Line 81: are the snow microstructure and density values used here single values or probability distributions? Are they determined according to the in-situ snowpit observations?

We presented probability distributions of microstructure and density values, but single values (mean values) were used in the final simulation. The distributions are shown so they can be used in future MCMC retrievals as priors.

This sentence was modified as follows.

*Secondly, we presented in situ measurements of snow microstructure and density in both main tundra snow layers (depth hoar and wind slab), mean ratios of layer thickness ==and the depth hoar fraction (DHF)== relative to snow depth.*
* * *
R2.C11 Figure 5(b) was not described in the caption.

The following was added in the caption.

[revised manuscript text omitted]